# Influence of the Antibiotic Oxytetracycline on the Morphometric Characteristics and Endophytic Bacterial Community of Lettuce (*Lactuca sativa* L.)

**DOI:** 10.3390/microorganisms11122828

**Published:** 2023-11-21

**Authors:** Natalia Danilova, Gulnaz Galieva, Polina Kuryntseva, Svetlana Selivanovskaya, Polina Galitskaya

**Affiliations:** Institute of Environmental Sciences, Kazan Federal University, Kazan 420008, Russia; natasha-danilova91@mail.ru (N.D.); goolnaz@rambler.ru (G.G.); polinazwerewa@yandex.ru (P.K.); svetlana.selivanovskaya@kpfu.ru (S.S.)

**Keywords:** antibiotics, antibiotic-resistant genes, endophytic bacteria, plant resistome, vegetable production, hydroponic cultivation

## Abstract

Antibiotics enter the soil with compost prepared from livestock manures and other sources. There is concern that they may influence plant growth and cause antibiotic resistance in soil and plant endospheric microbiomes. In the present work, lettuce plants were cultivated in soil and hydroponics spiked with oxytetracycline (0, 15, and 300 mg × kg^−1^ and 0, 15, and 50 mg × L^–1^, respectively) during a 28-day greenhouse experiment. It was revealed that the antibiotic reduced the chlorophyll content, the biomass, and the length of the roots and stems by 1.4–4.7, 1.8–39, 2.5–3.2, and 1.8–6.3 times in soil and in hydroponics. The copy numbers of the *tet*(A) and *tet*(X) genes were revealed to be 4.51 × 10^3^–1.58 × 10^5^ and 8.36 × 10^6^–1.07 × 10^8^ copies × g^–1^, respectively, suggesting the potential migration of these genes from soil/hydroponics to plant roots and leaves. According to a non-metric multidimensional scaling (NMDS) analysis of the 16S rRNA amplicon sequencing data, endospheric bacterial communities were similar in leaves and roots independent of the growing substrate and antibiotic concentration. While soil bacterial communities were unaffected by the presence of antibiotics, hydroponic communities exhibited dependency, likely attributable to the absence of the mitigating effect of soil particle absorption.

## 1. Introduction

Antibiotics and bacterial resistance to them are a global problem today, as they are considered new environmental pollutants [1]. The livestock sector is making a negative contribution to this situation since antibiotics are used to treat and prevent animal diseases and to accelerate their growth [2,3,4].

Abundant quantities of different groups of antibiotics are found in manure, soils, wastewater, and water reservoirs. In addition to the antibiotics themselves, antibiotic-resistant genes are found in these environments [5]. Soil is a huge reservoir of antibiotic-resistant bacteria. The main source of their entry into the soil is fertilization with manure and compost, as well as irrigation with wastewater [6,7]. In some countries of the world, tetracyclines are actively used in veterinary medicine and to accelerate the growth of animals [8]. According to 2020 data, 33,305 tons of tetracycline antibiotics were used in livestock breeding [9]. Up to 70–90% of antibiotics end up in manure unchanged. Indeed, during the storage or biological processing of manure, most of these antibiotics decompose, but the communities of composts, manures, and subsequently treated soils contain significant levels of antibiotic-resistant genes. In recent years, the content of genes resistant to antibiotics such as tetracyclines and sulfonamides has increased in the soil [10].

Plants are thought to have their own microflora distinct from that of the soil, but there is concern that the vertical migration of bacteria from the soil to crop plants may still occur [11,12]. Early studies have shown that the fertilization of agricultural soils with manure containing antibiotic-resistant bacteria can facilitate their spread in crops and their further movement through the food chain to people [1,4,13]. The soil rhizosphere is a hotspot where the active spread and transfer of antibiotic-resistant genes occurs because of the release of exudate from plant roots [3,14,15]. Antibiotic-resistant bacteria, including pathogens, can be found in roots (root endophytes) and leaves (leaf endophytes) and located on the surface of leaves and stems (phyllospheric microorganisms). Of particular concern are endophytic antibiotic-resistant bacteria because they cannot be removed from crops by washing them with water [3].

Hydroponically grown plant products are now very popular, as they allow for the efficient use of soil and water and achieve high yields and economic benefits. Recently, the use of purified wastewater was introduced to grow plants hydroponically. It is believed that products grown hydroponically meet all sanitary rules and are free from microbiological contamination. However, evidence has been obtained of the presence of antibiotic-resistant bacteria in the water of hydroponic systems, as well as in plants grown using hydroponics. However, data on the transfer of antibiotic-resistant genes into plants and, specifically, into the plant microbial community remain scarce. Therefore, there is now growing concern about the consumption of such products, as it concerns people’s health when treated with antibiotics, whose therapeutic effectiveness may be reduced [13].

Antibiotics, by negatively affecting the functioning of soil microflora, can indirectly affect the physiology of the plant itself. Being in the soil, antibiotics can have a depressing effect on the bacterial community, in particular, the rhizosphere microbiome, causing disturbances in nitrogen metabolism, which can seriously affect the growth and development of plants. The effect of antibiotics on plant physiology may depend on their dosage in the soil and stimulate growth at low concentrations, while at high concentrations, they can have toxic effects [16,17]. First of all, antibiotics negatively affect the length and elongation of roots and the number of lateral roots [18,19]. In addition, the presence of antibiotics can negatively affect plant biomass, shoot length, number of leaves, photosynthesis and transpiration rates, chlorophyll content, stomatal function, and chloroplast–synthase activity [18,20,21]. Thus, the use of antibiotics raises concerns in terms of reducing crop yields and exacerbating the issue of food security.

The aim of this work was (1) to analyze the effect of different concentrations of the antibiotic oxytetracycline (OTC) on the morphometric characteristics, biomass, and content of chlorophyll of lettuce (*Lactuca sativa* L.) and (2) to evaluate the vertical migration of the tetracycline-resistant genes *tet*(A) and *tet*(X) from soil and hydroponic system to the roots and leaves of lettuce. The antibiotic resistance of the soil and hydroponic bacteria was encouraged by the addition of OTC to soil and hydroponics in concentrations of 15 to 300 mg × kg^−1^ and 15 to 50 mg × L^−1^, respectively. The hydroponic experiment was considered in our study as a model one, since hydroponics, unlike soil, is a more homogeneous substrate, and all processes and mechanisms associated with antibiotic resistance can be more clearly understood in a hydroponic experiment. The antibiotic was used not as a factor reflecting environmental conditions but as a factor to model the presence of antibiotic-resistant species in the soil.

## 2. Materials and Methods

### 2.1. Hydroponic Experiment Design

Lettuce was used for the hydroponic experiment. Lettuce seeds were preliminarily germinated on vermiculite for 10 days. After that, seedlings in the amount of 9 pieces were transplanted into containers with a hydroponic mixture and grown for 28 days at a room temperature of 25 °C.

The hydroponic system consisted of the following macronutrients—N–NO_3_ (14.0 mM), N–NH^4+^ (2.0 mM), P (2.0 mM), K (10.0 mM), Ca (4.5 mM), Mg (2.0 mM), S–SO_4_ (5.0 mM) [22]. The following salts were used to prepare the hydroponic solution: 5[Ca(NO_3_)_2_ × 2H_2_O], NH_4_NO_3_, KH_2_PO_4_, MgSO_4_ × 7H_2_O, KNO_3_, K_2_SO_4_. The volume of the hydroponic mixture in each container was 12 L. Every 7 days, the hydroponic mixtures were replaced with fresh ones. OTC was used as a model antibiotic, which was added to the test systems in two concentrations: 15 mg × L^−1^ and 50 mg × L^−1^. Accordingly, 3 types of mixture were studied: without the addition of OTC (control), with 15 mg × L^−1^ of OTC, and with 50 mg × L^−1^ of OTC. 

Lettuce plants were harvested after 28 days of cultivation; leaves and roots were washed with water and dried on paper. Each plant was measured for chlorophyll content, biomass, root, and stem length, some of which were immediately cut off from the main plant for subsequent DNA extraction. Prior to DNA isolation, the selected roots and leaves were stored at −80 °C. The chlorophyll content in the lettuce was measured using Dualex Scientific+™ (Force-A, Orsay, France).

### 2.2. Soil Experiment Design

Soil poor in organic matter content was used for the soil experiment. Soil samples were taken at a depth of 20–40 cm in the backyard of the Kazan Federal University main building (55.790934, 49.120356). Lettuce was used as the test plant. Lettuce was grown in 38 × 26 cm plastic containers filled with 10 kg of soil. Before the experiment, the soil was moistened to 60% of the total moisture capacity and pro-incubated for 10 days. Seeds of lettuce were pre-germinated on vermiculite for 10 days [22]. Next, lettuce plants were transplanted into containers with soil in the amount of 9 pieces per container.

To provide plants with nutrition, at the beginning of the vegetation experiment, mineral fertilizers were applied to the soil once: superphosphate in the amount of 1.33 g × kg^−1^ and potassium nitrate in the amount of 3.5 g × kg^−1^. The antibiotic was introduced to the soil in two concentrations—15 mg × kg^−1^ and 300 mg × kg^−1^ [2,23]. The vegetation experiment was carried out in a greenhouse with a light regime of 16:8 h and a temperature of 22 °C. The duration of the experiment was 28 days [22]. On the 28th day, the lettuce plants were removed from the soil; the roots were washed and dried; and the morphometric data, biomass, and chlorophyll content of the plants were determined. A portion of the leaves and roots of the lettuce plants from each container was immediately used to extract bacterial DNA. The chlorophyll content in the lettuce was measured using Dualex Scientific+™ (Force-A, Orsay, France).

### 2.3. DNA Extraction, qPCR, and Sequencing

Extraction of DNA from plant and soil samples was carried out using the FastDNA Spin Kit for Soil (MP Bio, Irvine, CA, USA) according to the manufacturer’s instructions. DNA purification was carried out using the QIAquick PCR Purification Kit (Qiagen, Düsseldorf, Germany).

Detection of the bacterial 16S rRNA gene copy number was carried out via real-time PCR using primers 947f/1349r with the following sequences: F: AACGCGAAGAACCTTAC, R: CGGTGTGTACAAGGCCCGGGAACG [24]. The master mix reaction mixture (25 μL) contained the following components: DNA template—1 μL, forward and reverse primers (10 μM)—0.5 μL each, dNTPs (10 μM)—2.5 μL, 10x buffer with SYBR green—2.5 μL, MgCl_2_ (25 μM)—2.5 μL, SynTaq polymerapse (5 U μL^−1^)—0.2 μL, and ddH_2_O—15.3 μL. Amplification was performed on a BioRad CFX–96 cycler (BioRad, Hercules, CA, USA) using the following temperature program: primary denaturation at 95 °C for 5 min and then 39 three-step cycles at 60–62 °C for 45 s at 95 °C for 15 s and at 72 °C for 30 s. DNA standard curves were constructed using the bacterium *Pseudomonas putida*.

Detection of *tet*(A) and *tet*(X) genes was carried out via real-time PCR using three pairs of primers with the sequences in parentheses for *tet*(A) (F: CAGGCAGGTGGATGAGGAA, R: GGCAGGCAGAGCAAGTAGAG) and *tet*(X) (F: GAAAGAGACAACGACCGAGAG, R: ACACCCATTGGTAAGGCTAAG) [24,25]. In detail, the PCR products of *tet*(A) and *tet*(X) were first cloned using the TA Cloning Kit (Invitrogen Corporation, Carlsbad, CA, USA). Then, the plasmids carrying *tet*(A) and *tet*(X) were extracted and purified using a PureLink Quick Plasmid Miniprep Kit (Invitrogen Corporation, Carlsbad, CA, USA). The master mix reaction mixture (25 μL) contained the following components: DNA template—1 μL, forward and reverse primers (10 μM)—0.5 μL each, dNTPs (10 μM)—2.5 μL, 10x buffer with SYBR green—2.5 μL, MgCl_2_ (25 μM)—2.5 μL, SynTaq polymerase (5 U μL^−1^)—0.2 μL, and ddH_2_O—15.3 μL. Amplification was performed on a BioRad CFX–96 cycler using the following temperature program for the *tet*(A) and *tet*(X) genes: primary denaturation at 95 °C for 5 min and then 39 three-step cycles at 60–62 °C for 45 s, at 95 °C for 15 s, and at 72 °C for 30 s.

Sequencing of the bacterial community was performed on the Illumina base (Illumina, San Diego, CA, USA). The preparation of the genomic library was carried out according to the 16S Metagenomic Sequencing Library Preparation Protocol (Illumina MiSeq, San Diego, CA, USA). Amplification of the V3–V4 region of the 16S rRNA was performed using a DNA Engine Tetrad^®^ 2 cycler thermal cycler (BioRad, Hercules, CA, USA) using specific primers: A (TCGTCGGCAGCGTCAGATGTGTGTATAAGAGACAGCCTACGGGNGGCWGCAG) and B (GTCTCGTGGGCTCGGAGATGTGTATAAGAGACAGGACTACHVGTATCTAATCC).

Amplification was carried out according to the following regime: 95 °C for 3 min; 27 cycles of 95 °C (30 s), 55 °C (30 s), and 72 °C (30 s); and a final elongation at 72 °C (3 min). Amplicons were purified using an Agencourt AMPure XP purification kit (Beckman Coulter, Brea, CA, USA). The second round was carried out using the same amplification parameters. Amplicon concentration was measured on a Qubit 3.0 Fluorometer (Invitrogen, Carslbad, CA, USA) using a Quant-iT™ dsDNA High-Sensitivity Assay Kit (Thermo Fisher, Waltham, MA, USA). The quality control of the obtained amplicons was carried out on a Labchip GX Touch 24 instrument (PerkinElmer, Waltham, MA, USA). Next, sequencing was carried out on a MiSeq instrument (Illumina, San Diego, CA, USA), according to the manufacturer’s instructions.

16S pRNA sequencing data were analyzed using the Quantitative Insights Into Microbial Ecology (QIIME, East Lansing, MI, USA) platform, version 1.6.0 [26]. Representative sequences were aligned according to the PyNAST algorithm [27]. The taxonomic structure was determined against the Greengenes database using the USEARCH program. The operational taxonomic units were grouped with a similarity threshold of 97%. 

### 2.4. Statistical Analysis

All measurements were carried out in at least five repetitions. The statistical processing of the obtained results was carried out using Microsoft Office Excel 2010 (Redmond, WA, USA). All graphical data contain average values and standard errors. To assess the significance of differences, the Mann–Whitney test was used at α = 0.05.

## 3. Results and Discussion

### 3.1. Hydroponic Experiment

The presence of antibiotics in the substrate (soil, hydroponic system, etc.) can have both a direct negative effect on the germination, growth, and development of plants and an indirect effect through the bacterial community of the substrate and the endophytic plant community associated with it [18,19]. Since OTC is a hydrophilic compound, we compared its effect on lettuce plants grown in two media of different homogeneity—a hydroponic system and soil. The bioavailability of the antibiotic and the severity of its negative effect depends on the type of medium [28].

In the first step, we studied the effect of OTC on lettuce grown in a hydroponic system. Figure 1 shows the results of assessing the morphometric parameters, biomass, chlorophyll content, and photographs of lettuce grown in a hydroponic system containing 15 and 50 mg × L^−1^ of OTC. In the presence of OTC in two variants of hydroponic systems, the morphometric parameters (the length of roots and stems and the biomass), and the chlorophyll content in lettuce were significantly lower (*p* < 0.05) than in the control variant without the addition of OTC. Compared with the control in samples with 15 and 50 mg × L^−1^ of OTC, the length of the roots was 2.5 and 2.8 times smaller, the length of the stems was 1.8 and 2.7 times smaller, the biomass was 5.9 and 17.7 times smaller, and the chlorophyll content was 2.9 and 4.7 times lower, respectively. A number of studies have also described the negative effect of antibiotics on the morphometric parameters of plants. For example, a decrease in plant biomass and number of leaves, a change in the root and stem length ratio, and a decrease in the length of stems have been demonstrated in the presence of tetracyclines [21,29,30]. It has been shown that, in the presence of sulfadimethoxine, there is a significant inhibition of the development of all plant organs, namely, a decrease in the length and number of roots, hypocotyls, cotyledons, cotyledon petioles, internodes, and leaves [17]. The above changes are based on the inhibition of phosphatase activity, the increased release of electrolytes from roots, the decreased activity of mitochondrial cytochrome c oxidase, the decreased activity of peroxidase, the decreased rate of photosynthesis, the transpiration and stomatal conductance, and the increased intercellular concentration of CO_2_ in the presence of antibiotics [19,20,21,31].

The effect of antibiotics on plants can also be indirect through a change in the number and diversity of the bacterial community of the rhizosphere, which is actively involved in the growth and development of plants. A change in the composition of the bacterial community of the rhizosphere and root endosphere was demonstrated in the presence of macrolides and sulfonamides [32]. 

Figure 2 shows the results of the assessment of the bacterial 16S rRNA gene copy number in the roots and leaves of lettuce grown in a hydroponic system containing 15 and 50 mg × L^−1^ of OTC, as well as in the water of the hydroponic system. It can be seen that both in the control and in the variants with the antibiotic, the bacterial 16S rRNA gene copy number in the leaves of lettuce was significantly higher than in the roots and water (*p* < 0.05). When the antibiotic was added, there was a slight increase in the number of bacterial gene copies in the endosphere of the lettuce leaves compared with the control, but the differences were not statistically significant. The growth in bacterial numbers in the presence of antibiotics may be associated with the unhindered proliferation of antibiotic-resistant species, lacking competition from non-resistant species [33]. 

The structure of bacterial communities in the lettuce endosphere and the hydroponics water was revealed on the basis of 16S rRNA amplicon sequencing on the Illumina platform. 

The alpha diversity of the endophytic bacterial community, as well as the community of the aquatic environment of the hydroponic system, was assessed using the Evenness index (E), which reflects the number of OTUs, and the Shannon and Simpson indices, which reflect the diversity and uniformity of species distribution [34].

According to the results of the assessment of all three indices, it was found that the diversity of bacteria in the water of the hydroponic system was significantly higher (*p* < 0.05) than in the leaves and roots of the lettuce (Figure 3a,b), both in the presence and in the absence of the antibiotic. Indeed, the bacterial community of the plant roots and leaves was formed mainly by several bacterial species that were present in the seed. After germination, these microorganisms form bonds with the bacterial community of the nutrient substrate on which the plant grows. Not all microorganisms from the substrate penetrate the plant because of the immune system of the plant and the competitive and symbiotic relationships between the microorganisms of the substrate and “native” microorganisms from the seed. As a result, the endophytic community of an adult plant consists of the species present in the seed and the species present in the substrate. A small proportion of the community is also occupied by accidentally penetrating species (for example, from the air through damage in plant cells) [32,35].

It was found that, in the water of the hydroponic system of the control variant, the bacterial diversity was significantly higher (*p* < 0.05) than in the presence of OTC, which indicates the direct selective effect of the antibiotic on bacteria—the elimination of unstable and the active reproduction of resistant species [36,37]. In contrast, the diversity of bacteria within the plant did not depend on (in the case of the Shannon and Simpson indices) or change less significantly (in the case of the E index) in the presence of the antibiotic. It is known that antibiotics (in particular, oxytetracycline) are able to penetrate plant tissues, but the penetration process lasts several days or weeks. Thus, in the work of Al-Rimawi et al., it was demonstrated that the maximum concentration of oxytetracycline in the leaves of citrus plants occurred 2 weeks after the start of watering the plants with an antibiotic solution [38]. The lack of effect of oxytetracycline on the diversity of endophytes, revealed in our work, may be due to both the low concentration and insufficient duration of the effect of the antibiotic and the antibiotic resistance of endophytic species.

More than forty tetracycline-resistant genes are described in the literature, and in our work, we used two of them—*tet*(A) and *tet*(X)—as they are the most frequently used by researchers and responsible for different mechanisms of bacterial resistance—the efflux pump system (*tet*(A)) and the degradation of antibiotics in the cell (*tet*(X)) [25]. Figure 4 shows that, in the control variant of the hydroponic system without the antibiotic, the *tet*(A) and *tet*(X) genes were not expected to be found either in the lettuce plants or in water. For hydroponic systems with the antibiotic, the *tet*(A) gene was detected in the leaves and roots of the lettuce at both concentrations of the antibiotic. The *tet*(X) gene was found in water and in roots, but only at a higher dose of 50 mg × L^−1^. It is known that the half-life of oxytetracycline in the aquatic environment is quite short—from 2 to 12 days [39,40]—which is probably why no species resistant to it with the *tet*(A) gene were found on the 28th day of the experiment. The preservation of the *tet*(X) genes that we discovered can be explained by lower energy costs for its transmission compared with the *tet*(A) gene [41].

Within the endophytic community, antibiotic-resistant genes were more present in roots than in leaves. Those genes might be horizontally transferred from the substrate or rhizospheric microorganisms and developed in response to antibiotic presence [42]. Both mechanisms are more likely in roots than in the leaves of plants.

In order to reveal which OTUs might be potential carriers of antibiotic resistance, we calculated the correlation coefficient between the relative amount of each OTU in all samples and the antibiotic-resistant gene copy number. In Table 1, OTUs with R > 0.8 are presented. Among these bacterial OTUs, there were pathogens (*Clostridium* spp., *Ochrobactrum* spp., family Enterobacteriaceae), phytopathogens (*Erwinia* spp., family Xanthomonadaceae), symbiont bacteria (*Rhizobium leguminosarum* spp.), and bacteria living in various environments—water, soil, and plants (family Bacillaceae, *Sphingobacterium multivorum*, *Bdellovibrio bacteriovorus*). It can be assumed that these OTUs were the carriers of these resistant genes.

### 3.2. Soil Experiment

Soil is a complex system since antibiotics can be sorbed on soil particles and then desorbed after some time as a result of physical and chemical processes. Therefore, the bioavailability of antibiotics in soil differs from that in an aquatic environment [55].

In the next step, we studied the effect of antibiotics on lettuce grown in soil. Figure 5 shows the results of assessing the morphometric parameters, biomass, chlorophyll content, and photographs of lettuce grown in soil containing the antibiotic at concentrations of 15 and 300 mg × kg^−1^. The biomass, length of roots and stems, and chlorophyll content were the highest for the control variant of the soil system and amounted to 11.9 g, 8.9 cm, 21.4 cm, and 10.9 µg × cm^2^, respectively. In general, the values of the morphometric parameters, biomass, and chlorophyll content in the control soil experiment were higher than in the control hydroponic experiment.

For soil systems containing the antibiotic, the morphometric parameters, biomass, and chlorophyll content were lower, except for the length of roots and stems in the sample with 15 mg × kg^−1^ of OTC. In this sample, the length of the roots and stems correlated with the control values. As in the case of hydroponic-grown lettuce, the presence of the antibiotic resulted in the inhibition of plant growth and development. The literature also describes the inhibition of various morphometric parameters of plants grown in soil contaminated with antibiotics or treated with manure with residual antibiotic content. [29,56,57].

Figure 6 shows the results of the bacterial 16S rRNA gene copy number assessment. It can be seen that, as in the case of lettuce grown under the conditions of the hydroponic system, in the control and in the variants with the antibiotic, the number of bacteria in the leaves of lettuce was significantly higher (*p* < 0.05) than in the roots and soil (*p* < 0.05). In the presence of the antibiotic, there was a slight decrease in the number of bacterial genes in the soil and in the endospheres of the leaves and roots in the soil with both doses of the antibiotic, but the differences were not statistically significant. The results obtained are probably associated with the elimination of antibiotic-sensitive bacterial species [58]. 

The results of the alpha diversity assessment using the Evidence, Simpson, and Shannon indices are shown in Figure 7a,b. The number of OTUs in the soil of the control variant was higher than that in the water of the control variant of the hydroponic experiment. However, in leaves and roots, the number of OTUs was similar for both soil and hydroponic experiments since the endophytic bacterial community of plants is determined mainly by the composition of the seed bacterial community [35]. It was found that the greatest diversity of the bacterial community was in the soil samples, and the least was in the samples of the leaves and roots of lettuce. The literature also describes a higher diversity of bacterial species in soil than in plants [32,59]. In contrast to the hydroponic experiment with lettuce, soil contamination with OTC did not lead to significant differences in the level of alpha diversity in the bacterial community between samples with different doses, which is probably due to the sorption of the antibiotic in the soil [60].

Figure 8a,b presents the results of assessing the number of tetracycline-resistant genes (*tet*(A) and *tet*(X)) in the soil and endosphere of the lettuce. In the control variant, these genes, as expected, were not found in either the soil or the lettuce. When the antibiotic was introduced to the soil, tetracycline-resistant genes were found both in it and in the endosphere (leaves and roots) of lettuce and at both concentrations of the antibiotic. It should be noted that we did not detect the *tet*(A) genes in the water of the hydroponic system; the *tet*(X) genes were absent in the leaf microflora at both concentrations and in the root microflora at a low concentration of OTC. It is likely that the sorption and further desorption of the antibiotic from the surface of the soil particles caused its lower concentrations in the initial period but contributed to a longer preservation than in the aquatic environment. Indeed, the half-life of OTC in soil is 10–79 days, which is 5–7 times longer than its half-life in the aquatic environment [61]. The greatest number of tetracycline-resistant genes was found in plant roots; a similar fact was noted for the *tet*(A) genes in the hydroponic system. The reasons for this concentration of resistant species in the roots are discussed above. In general, the obtained data are consistent with the results of other researchers. Thus, a 160-fold increase was found in the amount of *sul1* gene in the bacterial community on the surface of lettuce grown in soil with manure [62]. Another study with lettuce grown in soil with manure showed an eight-fold increase in antibiotic-resistant genes [63].

As for the hydroponic system, we made an attempt to reveal the potential carriers of antibiotic-resistant genes in the soil and plant endophytic microbiome. A correlation was found between only one bacterial OTU (*Streptomyces* spp.) and the *tet*(A) resistant gene (correlation coefficient = 0.83). It can be assumed that this OTU was the carrier of this resistant gene.

### 3.3. Bacterial Community Analysis

#### Endospheric Bacterial Communities of Lettuce Plants

In the next stage, we compared the bacterial diversity of the endospheric bacterial community of lettuce grown under different conditions—soil and a hydroponic system. Despite the fact that soil bacterial communities are quite stable and do not change rapidly over time, antibiotic resistance properties are acquired and lost very quickly.

The total number of qualified sequences was 2,517,779 from all samples for the bacterial microbial community. The average number of sequences was 37,863, 63,605, and 44,571 for the water, soil, and plant (root and leaf) samples, respectively. More detailed information on sequencing quality is provided in Appendix A. Table 2 shows the dominant bacterial OTUs in the roots and leaves of the lettuce. It was found that, in the control samples, the dominant species in different parts of the lettuce were similar regardless of the method of cultivation, whether in hydroponics or in soil. Indeed, it has been shown that the main number of microorganisms is formed in plant organs from the seed, and a smaller number is introduced from the environment [35].

In the presence of OTC, changes in the number of dominant bacteria were found. Thus, in the hydroponic experiment, compared with the control, the abundance of bacteria of the orders Burkholderiales (family Oxalobacteraceae) and Methylophilales in the roots and the number of bacteria of the order Rhizobiales and class TM7-3 in the leaves decreased. Also, the presence of the antibiotic led to an increase in the number of bacteria of the orders Sphingomonadales, Bdellovibrionales (*Bdellovibrio bacteriovorus* spp.), Pseudomonadales, and Xanthomonadales. Moreover, for the last two orders of bacteria (Pseudomonadales and Xanthomonadales), the increase was the most significant—by 9–15 and 1.3–54 times compared with the control. In the soil experiment, the antibiotic caused a decrease in the number of bacteria of the Flavobacteriales (in roots) and Burkholderiales (in roots and leaves) orders and an increase in the number of bacteria of the Actinomycetales (in leaves), Sphingobacteriales (in leaves), and Pseudomonadales (in roots and leaves) orders. 

Interestingly, in the hydroponic experiment, the antibiotic caused more significant changes in the number of dominant bacteria than in the soil experiment. The bioavailability of the antibiotic for plants and its effect on the bacterial community were probably greater in the hydroponic experiment, where it was in the form of a solution, than in the soil experiment, where the medium was less homogeneous. Moreover, in the hydroponic experiment, differences were observed in the number of dominant bacteria between the root samples and leaf samples of lettuce. The less significant effect of the antibiotic on the bacterial community of the leaves can be explained by the fact that the roots and not the leaves were in direct contact with the water containing the antibiotic. The results obtained are in line with those presented in the scientific literature. Indeed, antibiotics caused alterations in endospheric bacterial communities’ structures [32,62,64], and those alterations were more pronounced in the root endosphere compared with the leaf endosphere [65,66].

In addition, we compared the dominant bacterial OTUs in the soil and water of the hydroponic systems in the presence of different antibiotic concentrations (Table 3). In general, soil and water differed in the set of dominant bacteria. Thus, in the soil, regardless of the concentration of the antibiotic, bacteria of the order Actinomycetales (Micrococcaceae and Nocardioidaceae families) had the highest amount.

In the hydroponic experiment, the list of dominant bacteria was different for the control and antibiotic variants. In the control sample, bacteria of the orders Actinomycetales (*Arthrobacter* spp.), Sphingomonadales (*Sphingomonas wittichii*), Rhizobiales (*Devosia* spp.), and Flavobacteriales were dominant, the abundance of which decreased significantly (*p* < 0.05) in the water samples after the addition of OTC. The introduction of OTC led to a significant increase (*p* < 0.05) in the abundance of bacteria of the orders Sphingobacteriales (*Sphingobacterium multivorum*), Bdellovibrionales (*Bdellovibrio bacteriovorus*), Rhizobiales (*Aminobacter* spp.), Pseudomonadales, and Methylophilales. These orders of bacteria were probably able to form resistance to OTC and actively multiply in the absence of competition.

In general, when the antibiotic was introduced, a more significant deviation in the number of dominant bacterial OTUs from the control was found in water than in soil. This is probably due to the fact that the OTC used in the experiment was water-soluble; therefore, in a liquid medium (water in the hydroponic experiment), its bioavailability turned out to be higher than in soil, where the antibiotic could be adsorbed by soil particles and not exert intense selective pressure on microorganisms [60].

The literature also describes the effect of antibiotics found in soil, even in residual concentrations, on the composition and structure of the bacterial community of rhizosphere soil and plants. Thus, it was revealed that a residual number of macrolides in the soil changed the composition of the bacterial community of the rhizosphere soil, and sulfonamides changed the composition of the bacterial community in the root samples [32].

Using NMDS analysis, similarities and differences were analyzed between the endospheric bacterial communities of the lettuce, as well as between the communities of the soil and hydroponic system in which it was grown (Figure 9). In general, the NMDS graph visualizes three groups of samples that differ from each other in diversity—(i) the endospheric bacterial community of the roots and leaves of the lettuce, (ii) the bacterial community of the water of the hydroponic experiment, and (iii) the bacterial community of the soil. It can be seen that the endospheric bacterial diversity of the roots and leaves of the lettuce turned out to be similar regardless of the location of its growth (soil and water). The introduction of the antibiotic at different concentrations did not lead to significant differences in the diversity of the bacterial community between the samples, with the exception of the root sample of the hydroponic experiment, where a higher concentration of OTC (50 mg × L^−1^) caused more significant changes in the number of dominant bacteria.

The diversity of the bacterial community of the soil samples was also similar regardless of the antibiotic concentration. The presence of OTC in the soil practically did not cause significant differences in the composition of the bacterial community between the soil samples containing different concentrations. The obtained data are consistent with the data presented in the literature describing shifts in the composition, structure, and diversity of the plant microbiome in the presence of antibiotics in soil [11,57,63,64,67,68]. Interestingly, the introduction of the antibiotic led to more significant differences in the diversity of the bacterial community between the water samples of the hydroponic experiment with different concentrations. As noted above, this may be due to a lack of mitigation on the part of the antibiotic, which was absorbed by soil particles.

## 4. Conclusions

In this work, it was found that the antibiotic oxytetracycline caused a dose-dependent reduction in the chlorophyll content, biomass, and morphometric characteristics of lettuce, regardless of its growth location (soil experiment/hydroponic experiment). For the hydroponic experiment, a change in the number of bacterial OTUs and the alpha diversity of the endospheric bacterial community of leaves and roots, as well as in the community in hydroponic water, was found. In the case of the soil experiment with lettuce, no effect of the antibiotic on the diversity of the bacterial community was revealed, probably because of the partial sorption of the antibiotic by soil particles. Regardless of the plant growth location, in the presence of OTC, the vertical migration of the *tet*(A) and *tet*(X) genes into the roots and further into the leaves was found.

## Figures and Tables

**Figure 1 microorganisms-11-02828-f001:**
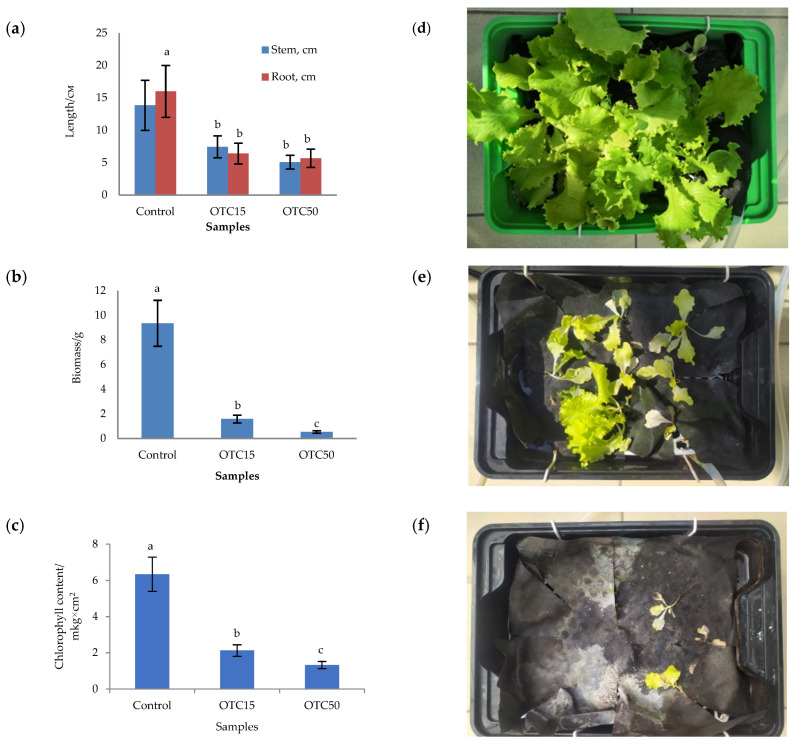
Length of stems and roots (**a**), biomass (**b**), and chlorophyll content (**c**) of lettuce in a hydroponic system containing OTC at concentrations of 15 and 50 mg × L^−1^ and photos of lettuce in the control sample (**d**), in a sample with 15 mg × L^−1^ of OTC (**e**), and in a sample with 50 mg × L^−1^ OTC (**f**). A difference significance analysis was calculated using the Mann–Whitney test. Different lowercase letters represent significant differences between different treatments (*p* < 0.05).

**Figure 2 microorganisms-11-02828-f002:**
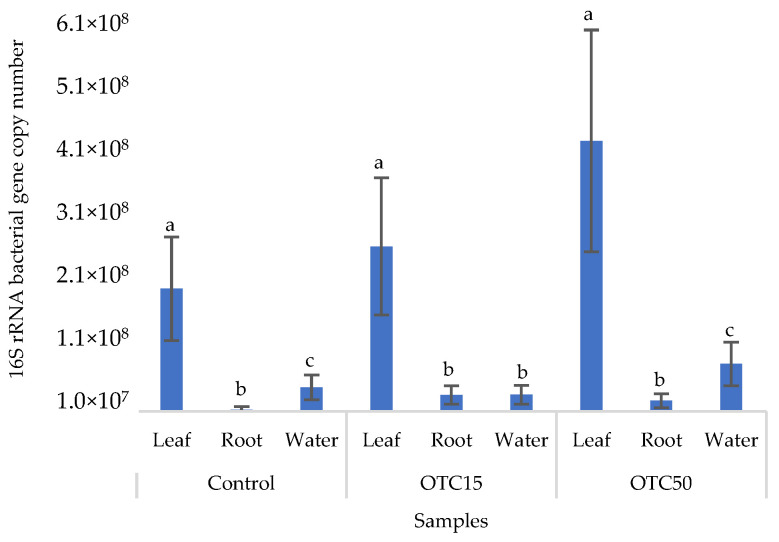
Bacterial 16S rRNA gene copy number in lettuce grown in a hydroponic system containing OTC at concentrations of 15 and 50 mg × L^−1^. A difference significance analysis was calculated using the Mann–Whitney test. Different lowercase letters represent significant differences between different treatments (*p* < 0.05).

**Figure 3 microorganisms-11-02828-f003:**
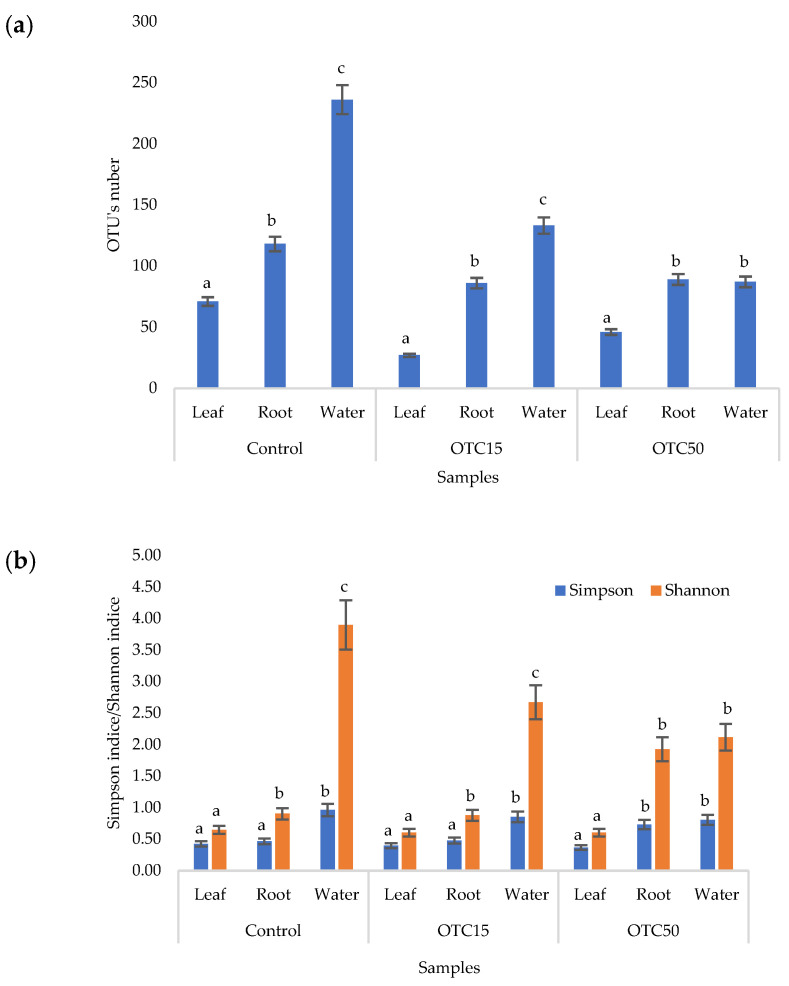
Number of bacterial OTUs (**a**) and Simpson and Shannon indices (**b**) of the bacterial community of lettuce grown in a hydroponic system containing OTUs at concentrations of 15 and 50 mg × L^−1^. A difference significance analysis was calculated using the Mann–Whitney test. Different lowercase letters represent significant differences between different treatments (*p* < 0.05).

**Figure 4 microorganisms-11-02828-f004:**
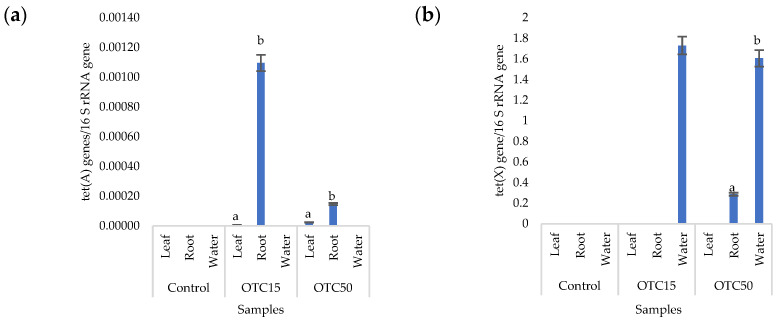
Number of *tet*(A) (**a**) and *tet*(X) (**b**) gene copies normalized to 16S rRNA bacterial gene copy numbers in lettuce grown in a hydroponic system containing OTC at concentrations of 15 and 50 mg × L^−1^. A difference significance analysis was calculated using the Mann–Whitney test. Different lowercase letters represent significant differences between different treatments (*p* < 0.05).

**Figure 5 microorganisms-11-02828-f005:**
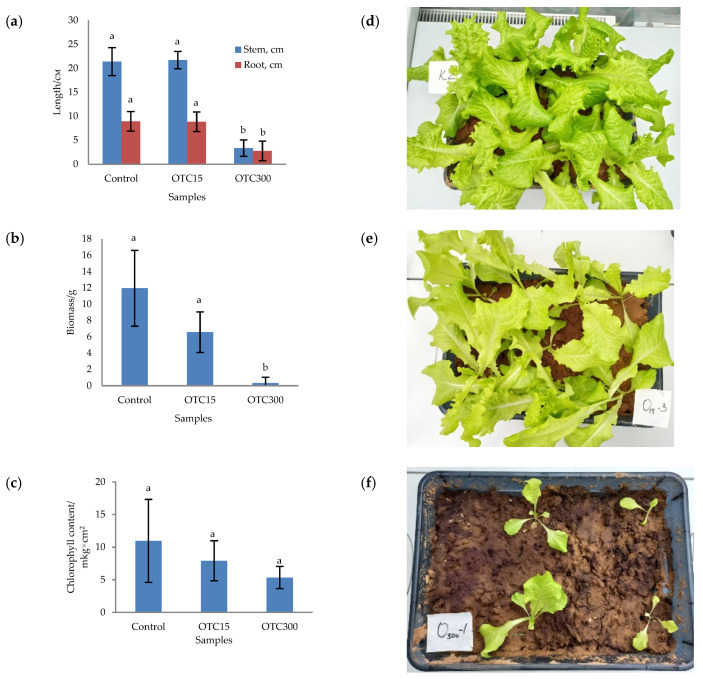
Length of stems and roots (**a**), biomass (**b**), and chlorophyll content (**c**) of lettuce grown in soil containing OTC at concentrations of 15 and 300 mg × kg^–1^ and photos of lettuce plants in the control sample (**d**), in a sample with 15 mg × kg^−1^ of OTC (**e**), and in a sample with 300 mg × kg^−1^ of OTC (**f**). A difference significance analysis was calculated using the Mann–Whitney test. Different lowercase letters represent significant differences between different treatments (*p* < 0.05).

**Figure 6 microorganisms-11-02828-f006:**
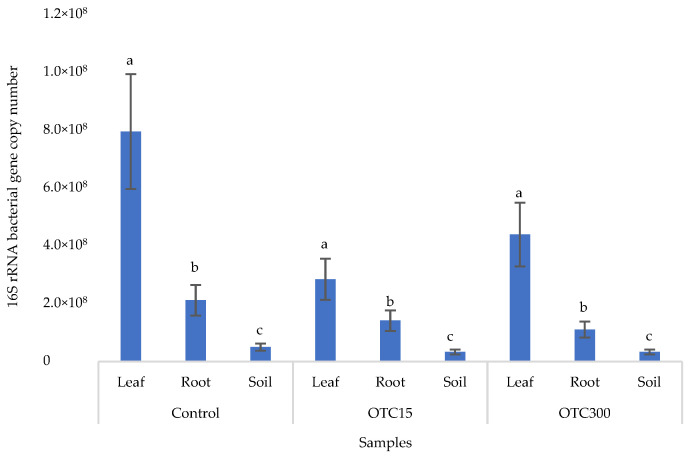
Bacterial 16S rRNA gene copy number in lettuce grown in soil containing OTC at concentrations of 15 and 300 mg × kg^−1^. A difference significance analysis was calculated using the Mann–Whitney test. Different lowercase letters represent significant differences between different treatments (*p* < 0.05).

**Figure 7 microorganisms-11-02828-f007:**
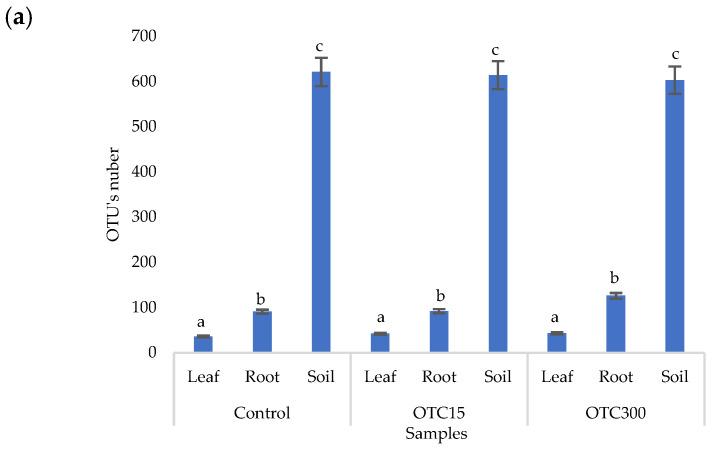
Number of bacterial OTUs (**a**) and Simpson and Shannon indices (**b**) of the bacterial community of lettuce grown in soil containing OTC at concentrations of 15 and 300 mg × kg^−1^. A difference significance analysis was calculated using the Mann–Whitney test. Different lowercase letters represent significant differences between different treatments (*p* < 0.05).

**Figure 8 microorganisms-11-02828-f008:**
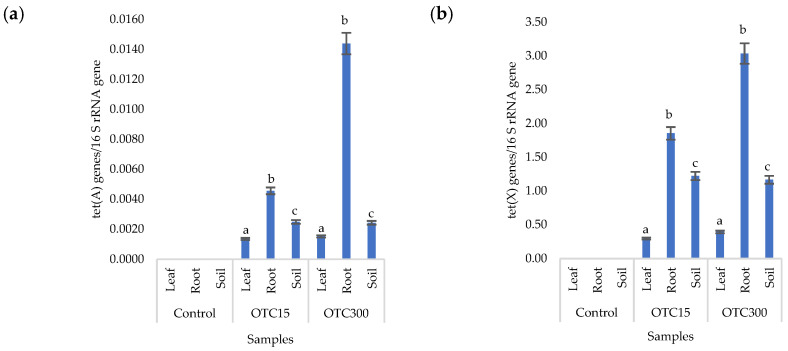
Numbers of *tet*(A) (**a**) and *tet*(X) (**b**) gene copies normalized to 16S rRNA bacterial gene copy numbers in lettuce grown in soil containing OTC at concentrations of 15 and 300 mg × kg^−1^. A difference significance analysis was calculated using the Mann–Whitney test. Different lowercase letters represent significant differences between different treatments (*p* < 0.05).

**Figure 9 microorganisms-11-02828-f009:**
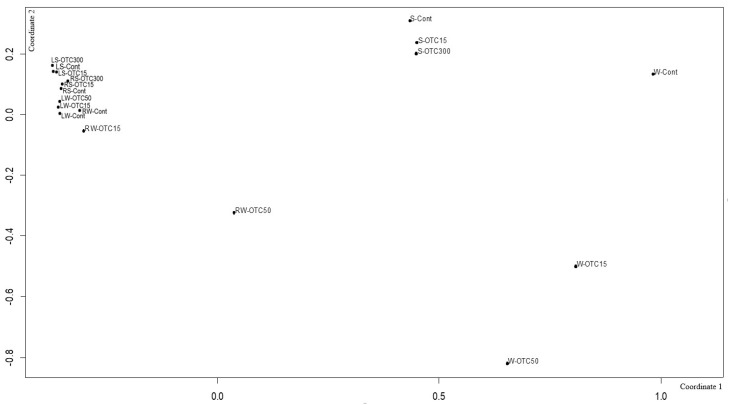
Beta diversity of the bacterial community of lettuce grown in soil without OTC and with OTC at concentrations of 15 and 300 mg × kg^−1^ (LS–Cont, LS–OTC15, LS–OTC300 and RS–Cont, RS–OTC15, RS–OTC300—leaves and roots of a soil experiment with 15 and 300 mg × kg^−1^ of OTC; LW–Cont, LW–OTC15, LW–OTC50 and RW–Cont, RW–OTC15, RW–OTC50—leaves and roots of a hydroponic experiment with 15 and 50 mg × L^−1^ of OTC; S–Cont, S–OTC15, and S–OTC300—soil from soil experiment; W–Cont, W–OTC15, and W–OTC50—water from hydroponic experiment).

**Table 1 microorganisms-11-02828-t001:** Correlation coefficient between the number of bacterial OTUs and the number of copies of the *tet*(A) and *tet*(X) genes in a hydroponic experiment with lettuce.

Bacterial OTU	Habitat/Short Description	Correlation Coefficient with *tet*(A) Gene Copies
f__Bacillaceae;g__;s__	Soil, air, aquatic ecosystems, composts, salt lakes [43].	0.86
g__Clostridium;s__	Soil; gastrointestinal tracts of animals and humans. There are several pathogenic species [44].	0.84
g__Ochrobactrum;s__	Water, soil, plants, and animals. Pathogen [45].	0.84
g__Rhizobium;s__leguminosarum	Formation of symbiosis with leguminous plants [46].	0.86
f__Enterobacteriaceae;Other;Other	Soil, aquatic ecosystems, animal and human gastrointestinal tracts, plants. There are several pathogenic species [47].	0.85
f__Enterobacteriaceae;g__;s__		0.83
g__Erwinia;s__	Phytopathogen [48].	0.85
f__Xanthomonadaceae;g__;s__	Phytopathogen [49].	0.92
g__Akkermansia;s__muciniphila	Human gastrointestinal tract [50].	0.85
**Bacterial OTU**	**Habitat/Short Description**	**Correlation Coefficient with *tet*(X) Gene Copies**
g__Pedobacter;s__	Soil [51]	0.81
g__Sphingobacterium;s__multvorum	Soil, aquatic ecosystems, plants, wastewater treatment plants [52].	0.92
g__Bradyrhizobium;Other	Nitrogen fixation [53].	0.90
g__Aminobacter;s__	Soil [53].	0.92
g__Mesorhizobium;Other	Nitrogen fixation [53].	0.81
g__Mesorhizobium;s__		0.98
g__Bdellovibrio;Other	Different habitats [54].	0.97
g__Bdellovibrio;s__bacteriovorus		0.93
f__Moraxellaceae;g__;s__	Aquatic ecosystems; soil [53].	0.92

**Table 2 microorganisms-11-02828-t002:** Number of dominant bacteria in **lettuce** grown in soil and under hydroponic experimental conditions.

Bacterial OTU	Hydroponic Experiment					
	Leaf			Root		
	Control	OTC15	OTC50	Control	OTC15	OTC50
Unassigned;Other;Other;Other;Other;Other;Other	54.3	73.5	42.4	7.2	9.1	1.4
f__Micrococcaceae;g__;s__	3.1	0.0	0.0	0.0	0.1	0.0
g__Rhodococcus;s__	0.0	0.0	0.0	0.0	0.0	0.0
f__Streptomycetaceae;g__;s__	0.0	0.0	0.0	0.0	0.0	0.0
g__Streptomyces;s__	0.0	0.0	0.0	0.2	0.0	0.0
g__Rubrobacter;s__	0.0	0.0	0.0	0.0	0.0	0.0
g__Flavobacterium;s__	0.0	0.0	0.0	0.3	0.4	0.1
o__Sphingobacteriales;f__;g__;s__	2.1	3.5	1.9	3.3	2.4	0.1
g__Sphingobacterium;s__faecium	0.0	0.8	0.0	0.0	0.4	3.4
g__Sphingobacterium;s__multivorum	0.0	0.0	1.3	0.1	0.6	4.7
o__[Saprospirales];f__;g__;s__	0.0	0.3	0.6	0.7	0.9	0.0
f__Clostridiaceae;Other;Other	0.0	0.0	0.0	3.4	0.1	0.0
f__Peptostreptococcaceae;g__;s__	0.0	0.4	1.9	3.1	0.0	0.0
f__Caulobacteraceae;g__;s__	0.3	1.0	0.0	0.4	3.2	2.9
o__Rhizobiales;f__;g__;s__	1.4	0.0	0.0	0.4	1.3	6.2
f__Methylobacteriaceae;g__;s__	0.3	0.4	0.6	0.1	0.4	0.1
g__Aminobacter;s__	0.3	0.7	0.6	0.1	1.6	3.2
g__Agrobacterium;s__	0.7	0.0	0.0	5.0	1.8	0.1
g__Rhizobium;s__leguminosarum	0.0	0.0	0.0	1.2	6.2	1.5
f__Rhodospirillaceae;g__;s__	2.9	2.0	0.6	0.6	0.6	0.1
g__Sphingomonas;s__	5.0	6.1	7.6	8.8	6.9	0.4
g__Sphingomonas;s__wittichii	0.6	0.0	1.9	4.8	0.4	0.0
g__Achromobacter;s__	0.0	0.0	0.0	0.1	3.2	6.0
f__Comamonadaceae;g__;s__	1.0	0.0	1.3	1.2	1.1	0.2
f__Oxalobacteraceae;g__;s__	0.5	0.0	0.0	11.5	0.4	0.0
f__Methylophilaceae;Other;Other	0.0	2.3	0.0	0.3	10.2	0.4
f__Methylophilaceae;g__;s__	0.4	0.0	0.6	17.2	1.5	0.0
g__Bdellovibrio;s__bacteriovorus	0.0	0.3	2.5	0.0	2.4	8.6
g__Helicobacter;s__pylori	0.2	0.4	3.2	0.4	0.2	0.0
g__Pseudomonas;s__	0.3	2.3	3.2	1.3	16.0	20.1
g__Pseudomonas;s__fragi	0.0	0.0	0.0	3.6	2.4	0.0
g__Stenotrophomonas;s__	0.0	0.0	1.3	0.5	2.7	26.0
c__TM7-3;o__;f__;g__;s__	6.2	0.0	0.6	0.3	0.0	0.0
Unassigned;Other;Other;Other;Other;Other;Other	0.0	0.4	0.4	0.1	0.4	0.2
f__Micrococcaceae;g__;s__	0.2	0.0	0.0	0.0	0.0	0.1
g__Rhodococcus;s__	6.0	5.6	13.2	7.1	7.6	7.1
f__Streptomycetaceae;g__;s__	0.0	0.0	0.4	1.7	0.8	3.2
g__Streptomyces;s__	0.0	0.0	0.0	0.7	0.7	4.8
g__Rubrobacter;s__	1.7	3.3	2.3	1.9	2.2	1.2
g__Flavobacterium;s__	0.0	0.0	0.2	5.2	2.3	0.3
o__Sphingobacteriales;f__;g__;s__	3.9	5.2	6.5	4.2	3.8	3.6
g__Sphingobacterium;s__faecium	0.0	0.0	0.0	0.0	0.0	0.0
g__Sphingobacterium;s__multivorum	0.0	0.0	0.0	0.1	0.0	0.0
o__[Saprospirales];f__;g__;s__	5.0	1.0	2.2	2.2	1.5	2.1
f__Clostridiaceae;Other;Other	0.0	0.0	0.0	0.0	0.0	0.0
f__Peptostreptococcaceae;g__;s__	0.0	0.0	0.0	0.0	0.0	0.0
f__Caulobacteraceae;g__;s__	2.1	1.5	2.3	2.8	1.0	1.4
o__Rhizobiales;f__;g__;s__	0.4	0.2	0.4	0.4	0.2	0.4
f__Methylobacteriaceae;g__;s__	1.5	1.6	0.6	1.7	3.0	0.5
g__Aminobacter;s__	0.0	0.0	0.0	0.0	0.0	0.0
g__Agrobacterium;s__	0.0	0.0	0.0	1.3	0.3	0.3
g__Rhizobium;s__leguminosarum	0.0	0.0	0.0	0.0	0.0	0.0
f__Rhodospirillaceae;g__;s__	3.1	2.8	1.7	1.3	2.1	1.7
g__Sphingomonas;s__	49.4	49.2	48.4	28.2	40.7	35.9
g__Sphingomonas;s__wittichii	0.0	0.0	0.0	0.0	0.0	0.0
g__Achromobacter;s__	0.0	0.0	0.0	0.1	0.0	0.0
f__Comamonadaceae;g__;s__	6.6	7.9	2.4	6.0	3.4	3.5
f__Oxalobacteraceae;g__;s__	0.0	0.0	0.0	0.1	0.0	0.1
f__Methylophilaceae;Other;Other	0.0	0.0	0.0	0.0	0.0	0.0
f__Methylophilaceae;g__;s__	0.0	0.0	0.0	3.3	0.5	0.0
g__Bdellovibrio;s__bacteriovorus	0.0	0.0	0.0	0.0	0.0	0.0
g__Helicobacter;s__pylori	0.0	0.0	0.0	0.0	0.0	0.0
g__Pseudomonas;s__	3.1	7.1	6.5	3.7	8.4	6.9
g__Pseudomonas;s__fragi	0.0	0.0	0.0	1.6	0.0	0.7
g__Stenotrophomonas;s__	0.0	0.0	0.0	0.0	0.1	0.3
c__TM7-3;o__;f__;g__;s__	0.0	0.0	0.0	0.0	0.1	0.0

**Table 3 microorganisms-11-02828-t003:** Number of dominant bacteria in soil and water of a hydroponic system in the presence of different antibiotic concentrations.

Bacterial OTU	Water			Soil		
	Control	OTC15	OTC50	Control	OTC15	OTC300
Unassigned;Other;Other;Other;Other;Other;Other	6	0	0	1	1	1
o__Acidimicrobiales;f__;g__;s__	0	0	0	3	2	2
g__Lentzea;Other	0	0	0	2	1	1
f__Micrococcaceae;g__;s__	1	0	0	6	6	6
g__Arthrobacter;s__	16	0	0	3	3	3
f__Nocardioidaceae;g__;s__	1	0	0	4	3	3
g__Patulibacter;s__	0	4	0	0	0	0
g__Flavobacterium;s__	5	0	0	1	1	1
g__Pedobacter;s__	0	0	3	0	0	0
g__Sphingobacterium;s__faecium	0	3	0	0	0	0
g__Sphingobacterium;s__multivorum	0	2	17	0	0	0
g__Candidatus Rhabdochlamydia;s__	4	0	0	0	0	0
g__Bacillus;s__	0	0	0	3	2	2
o__Rhizobiales;f__;g__;s__	2	2	1	1	1	1
g__Devosia;s__	6	1	1	1	1	1
g__Aminobacter;s__	0	16	15	1	2	2
g__Agrobacterium;s__	3	0	0	0	0	0
f__Rhodospirillaceae;g__;s__	3	0	0	0	0	1
o__Rickettsiales;f__;g__;s__	4	0	0	0	0	0
g__Kaistobacter;s__	0	0	0	2	2	2
g__Novosphingobium;s__	2	0	0	0	0	0
g__Sphingobium;s__	4	0	0	0	0	0
g__Sphingomonas;s__wittichii	7	0	0	0	0	0
f__Oxalobacteraceae;g__;s__	1	0	0	2	2	3
f__Methylophilaceae;Other;Other	0	8	4	0	0	0
f__Methylophilaceae;g__;s__	2	1	0	1	2	2
g__Methyloversatilis;s__	0	5	0	0	0	0
g__Bdellovibrio;s__bacteriovorus	0	32	34	0	0	0
f__Coxiellaceae;g__;s__	2	0	0	0	0	0
g__Pseudomonas;s__	1	8	10	0	1	2
f__Xanthomonadaceae;g__;s__	0	0	0	3	3	2
g__Stenotrophomonas;s__	0	0	3	0	0	0
c__TM7-3;o__;f__;g__;s__	3	0	0	0	0	1

## Data Availability

Data are contained within the article and Appendix A.

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
