# Peer review of "Influence of the Antibiotic Oxytetracycline on the Morphometric Characteristics and Endophytic Bacterial Community of Lettuce (Lactuca sativa L.)"

_microorganisms, 2023, doi:10.3390/microorganisms11122828_

Round 1
Reviewer 1 Report (Previous Reviewer 1)
Comments and Suggestions for Authors
Horizontal and vertical coordinate units need to be further standardized.
For example, Figure1, C, what does the means of “Chlorophyll/mkg×cm2” ? The upper corner mark?
The note of Figure.1 (p<0,05), p should be Italic, and should be p<0.05?
Figure.2 6.10E+08 ect., is the right generic representation?
Figure.3 The horizontal and vertical coordinates should be specified
Figure.9 Beta diversity can be show by NMDS figure.
Table 3. and Table 4 should be changed to the three-line tabel.
Author Response
Please see the attachment.

Reviewer 2 Report (Previous Reviewer 3)
Comments and Suggestions for Authors
This version is much better. Still, the references need further revision according to the requirement of the journal. Please carefully check them.
Author Response
Please see the attachment.

Reviewer 3 Report (Previous Reviewer 4)
Comments and Suggestions for Authors
The paper is now in a very good shape.
I only noted that it is not necessary to use the Latin name including the authority of the plant so many times and the reference list should be formatted according to the journal style. Some errors have been pointed out in the text of the attached pdf..

Comments on the Quality of English LanguageAuthor Response
Please see the attachment.

Round 2
Reviewer 1 Report (Previous Reviewer 1)
Comments and Suggestions for Authors
accept
This manuscript is a resubmission of an earlier submission. The following is a list of the peer review reports and author responses from that submission.
Round 1
Reviewer 1 Report
Comments and Suggestions for Authors
1. Not innovative enough.
2. Hydroponic lettuce is one of the main cultivation methods at present,antibiotics occur in soils, soil is a very complex system, with a strong buffer capacity.
3. It is of little significance to study antibiotic experiment with lettuce hydroponics.
4. Soil microbial community is a relatively stable system, the study of soil microbial community structure changes in just 28 days is not convincing, not to mention endophytes.
5. The high-throughput sequencing data were not further analyzed and the quality of the charts was unsatisfactory.
Reviewer 2 Report
Comments and Suggestions for Authors
The manuscript entitled “Influence of the antibiotic oxytetracycline on the morphometric characteristics and endophytic microbial community of lettuce (Latuca sativa L.)” submitted by Danilova et al. analyzed the effect of oxytetracycline on chlorophyll content, morphometric characteristics, and biomass of lettuce, and also evaluated the vertical migration of tet (A) and tet (X) genes from soil and hydroponic system containing oxytetracycline to the roots and leaves of lettuce (Latuca sativa L.). The topic of the manuscript does fit the scope of the journal well. The paper is clearly structured and well organized. There are some problems need to be solved. I marked them on the attached file. In addition, the styles of the text and references should be checked carefully.

Comments on the Quality of English LanguageThis manuscript "Influence of the antibiotic oxytetracycline on the morphometric characteristics and endophytic microbial community of lettuce (Latuca sativa L.)" is an interesting piece of work worth publishing in Microorganisms but it needs revisions before it is accepted.
Reviewer 3 Report
Comments and Suggestions for Authors
Article "Influence of the antibiotic oxytetracycline on the morphometric characteristics and endophytic microbial community of lettuce (Latuca sativa L.)" study has been well planned and performed.
1. Can you add the difference significance analysis in Fig 1?
2. How do you explain that some of the bar is very long like in Fig 1, 2, 6?
3. Table 2 is not necessary. This information can just be included in the text.
4. Some writings are wrong. Like :
Delete the “.” At the end of the sentence in L28, L35
“μl” should be “μL”
Please correct carefully according to the requirements first.
Reviewer 4 Report
Comments and Suggestions for Authors
This paper describes a study based on the effects of an antibiotic that is mainly being used in human and veterinary medicine on a single plant. The study claims that it would investigate "endophytes", which is why I initially agreed to do the review.
However, most of the organisms that are normally present in a given plant are fungi, which were not even studied. Furthermore, not even the bacteria were isolated and characterized by decent methods, and the antibiotic is not likely to play any major role in the actual enviroment where the plants grow in nature.Tetracyclins are not brought out to the field (even though they are occasionally used in veterinary medicine), and their clinical use is very limited today. They are produced by some soil inhabiting actinobacteria, but this organnism group produced hundreds of different antibiotic compunds and most of their species are not closely associated with plants.
These facts preclude me from even asking for major revision. According to my experience the compositon of the endophytic microbiota and mycobiota (NOT: microflora because the latter outdated term refers to -flora = plants!) is highly dependent on the situation and the experiments of this study can hardly be taken seriously.
Comments on the Quality of English LanguageN/A